# LESS: Label-Efficient and Single-Stage Referring 3D Segmentation

**Xuexun Liu**[1*]**, Xiaoxu Xu**[1*]**, Jinlong Li**[2*]**, Qiudan Zhang**[1]**, Xu Wang**[1†]**, Nicu Sebe**[2]**, Lin Ma**[3]

[1]College of Computer Science and Software Engineering, Shenzhen University,
Shenzhen, 518060, China.
[2]University of Trento, Italy.
[3]Meituan Inc., China.

## Abstract

Referring 3D Segmentation is a visual-language task that segments all points of the specified object from a 3D point cloud described by a sentence of query. Previous works perform a two-stage paradigm, first conducting language-agnostic instance segmentation then matching with given text query. However, the semantic concepts from text query and visual cues are separately interacted during the training, and both instance and semantic labels for each object are required, which is time consuming and human-labor intensive. To mitigate these issues, we propose a novel Referring 3D Segmentation pipeline, **L**abel-**E**fficient and **S**ingle-**S**tage, dubbed **LESS**, which is only under the supervision of efficient binary mask. Specifically, we design a Point-Word Cross-Modal Alignment module for aligning the fine-grained features of points and textual embedding. Query Mask Predictor module and Query-Sentence Alignment module are introduced for coarse-grained alignment between masks and query. Furthermore, we propose an area regularization loss, which coarsely reduces irrelevant background predictions on a large scale. Besides, a point-to-point contrastive loss is proposed concentrating on distinguishing points with subtly similar features. Through extensive experiments, we achieve state-of-the-art performance on ScanRefer dataset by surpassing the previous methods about 3.7% mIoU using only binary labels. Code is available at `https://github.com/mellody11/LESS`.

## 1 Introduction

Referring 3D Segmentation task aims to segment the specific object from a 3D point cloud scene with a free-form natural language expression. It allows users to interact and analyze 3D data through verbal instructions or queries. This approach is particularly beneficial in applications that necessitate direct interaction with 3D environments, such as augmented reality (AR) systems, embodied-AI, robotics, virtual reality (VR) environments, and in fields like architecture and medical imaging, where precise identification and segmentation of objects based on descriptive queries can significantly enhance user experience and operational efficiency.

Previous Referring 3D Segmentation [13; 29] mainly leverage a two-stage workaround, as shown in Fig.1 (a). They typically adopt a 3D instance semantic segmentation network to get the instance proposals at the first stage. The predicted instance proposals will be utilized to match with the queries and finally get the final prediction mask according to the matching score. Although those method have achieved remarkable performance, there still exist some problems. First, owing to the large-scale and

---

[*]Equal contributions.
[†]Corresponding author.

38th Conference on Neural Information Processing Systems (NeurIPS 2024).

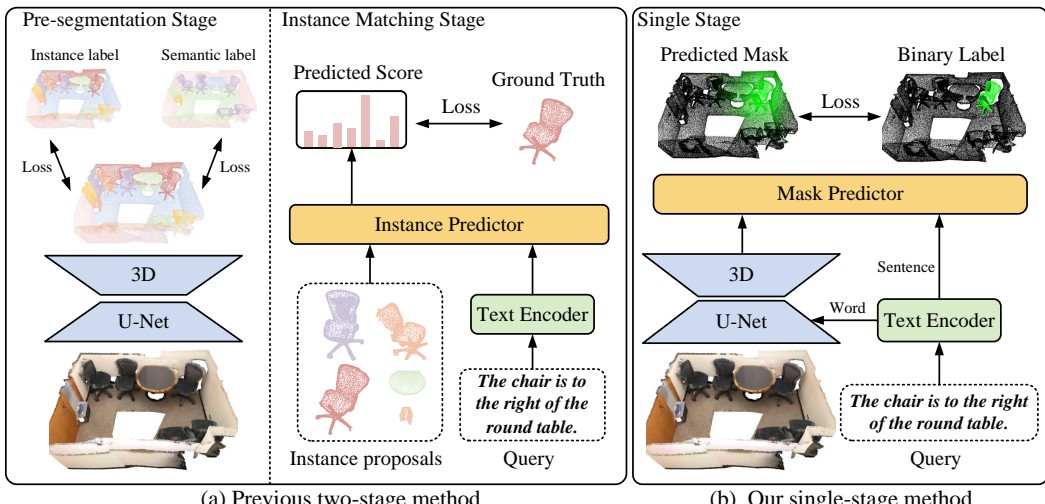

| Pre-segmentation Stage | Instance Matching Stage | Single Stage |

(a) Previous two-stage method    (b) Our single-stage method

Figure 1: Comparison between the two-stage method and our single-stage method. (a) The two-stage method initially performs instance segmentation with instance labels then semantic labels to get the instance proposals and bases on the provided query to match the most relevant instance proposal. (b) Our single-stage method only utilizes the binary mask of the described object for training and integrates language and vision features during feature extraction.

irregular 3D point clouds, some instance proposals may leave out the target in the pre-segmentation stage. Besides, lacking of linguistic guidance in the segmentation stage fails to focus on the objects that are more essential to the referring task. Moreover, existing Referring 3D Segmentation utilizes both instance labels and semantic labels to segment target proposal rather than binary mask used in referring image segmentation, which is more time consuming and labor intensive.

To address the aforementioned problems, we propose a **L**abel-**E**fficient and **S**ingle-**S**tage Referring 3D Segmentation method, namely **LESS**, which is under the supervision of binary mask, as shown in Fig.1 (b). We first process the query with text encoder to get the word-level feature and sentence-level feature. Then we extract the multi-modal feature with the guidance of text feature through a 3D sparse U-Net. Finally, the mask predictor aligns multi-modal features with language features, and directly predicts the mask of the described object. Here only object mask serves as the label to supervise the whole training procedure.

However, 3D point cloud inherently provide a higher level of complexity and a large scale. There exists numerous different objects in a single 3D scene compared to the referring image segmentation task. Besides, binary mask has less semantic meanings compared to instance labels and semantic labels. These challenges make it difficult to supervise our model to localize and segment target objects with only binary mask. Therefore, we propose to alleviate these problems by some ways, as shown in Fig.2. Firstly, to facilitate fine-grained alignment between points and words, we propose Point-Word Cross-Modal Alignment module. The PWCA module utilizes cross-modal attention in the multi-modal context to align textual features extracted by the text encoder with point cloud features, followed by further extraction of useful multi-modal information using robust 3D sparse convolutional layers. Thus we can extract a more semantic meaning fused feature. Meanwhile, we employ Query Mask Predictor, which utilizes the extracted multi-modal features to decode learnable query embeddings, generating candidate masks. By introducing the Query-Sentence Alignment modules, we can compute similarity between the decoded query embeddings and sentence features, using the similarity as weights to perform weighted summation of the candidate masks to produce the final mask. To address the significant interference caused by multiple objects and backgrounds, we propose an area regularization loss and a point-to-point contrastive loss. Area regularization loss reduces irrelevant background predictions by constraining the probabilities of the predicted mask, while point-to-point contrastive loss constrains the distances between positive and negative points in the latent space to achieve better segmentation.

To summarize, our contributions are as follows:

- We propose a new Referring 3D Segmentation method LESS, which directly performs Referring 3D Segmentation at a single stage to bridge the gap between detection and matching under the supervision of binary mask. To the best of our knowledge, LESS is the first work investigating label-efficient and single-stage in Referring 3D Segmentation task.

- Our LESS utilize a Point-Word Cross-Modal Alignment module to align fined-grained point and word features. Besides, we employed Query Mask Predictor module and Query-Sentence Alignment modules for coarse-grained alignment between masks and sentences. Moreover, the area regularization loss and the point-to-point contrastive loss are introduced to better support to eliminate interference caused by multiple objects and backgrounds.

- Extensive experiments confirm the effectiveness of our method. Our method outperforms the existing state-of-the-art method on ScanRefer dataset with only the supervision of binary mask. Our LESS and its results provide valuable insights to improve further research of label-efficient and single-stage Referring 3D Segmentation.

## 2 Related Works

### 2.1 Referring 3D Segmentation

Referring 3D Segmentation has previously received limited exploration, however, with the advancements in multi-modal learning and embodied AI, it is set to attract increasing interest in the future. TGNN [13] is the first to introduce Referring 3D Segmentation task. They initially trained an instance segmentation network, followed by a Graph Neural Network (GNN) to learn features of instances and their relationships guided by linguistic information. Building on TGNN, X-RefSeg [29] developed a cross-modal graph. They employed an GNN to model the texture and spatial relationships of instances, and refining the results through inference and matching processes.

### 2.2 3D Visual Grounding

3D visual grounding aims to locate the object within point clouds mentioned by free-form natural language descriptions. Most methods follow a two-stage detection-then-matching pipeline. Initially, they utilize a pre-trained 3D detector [28; 23] or segmenter [15; 33] to extract object representations. Subsequently, these methods align text features with object features to identify the best-matched object. Researchers primarily concentrate on the second stage which involves modeling object relationships and exploring feature fusion between language and objects. Methods employed include multi-modal feature concatenation [3; 1], attention-based multi-modal feature alignment [44; 11], graph neural network-based reasoning [13; 41], and the alignment of visual and language features aided by 2D images [40; 42; 38; 37].Other researchers have investigated single-stage approaches for 3D visual grounding. 3D-SPS [25] views the task as key point selection, progressively identifying keypoints with the guidance of language and directly locates the target. BUTD-DETR [14] employs a transformer decoder [2] to identify described objects using language cues and proposal boxes. Building on this, EDA [36] enhances dense alignment between objects and point clouds by explicitly decoupling textual attributes from sentences.

The distinction between 3D visual grounding and Referring 3D Segmentation lies in the latter's enhanced localization precision, offering significant value in applications like robotic grasping. While traditional single-stage 3D visual grounding methods regress a 3D bounding box, our single-stage approach decodes a binary mask for the entire point cloud scene, presenting a more complex challenge.

### 2.3 Referring Image Segmentation

Referring image segmentation is a visual task that involves pixel-level segmentation of an image's target object based on a referring expression. Early approaches [12; 20; 18; 4; 17] utilize CNNs and RNNs to extract features and then perform simple concatenation for multi-modal feature fusion. Subsequent research [16; 19; 9] adopt transformer models for more effective feature extraction and fusion. Recent studies [39; 10; 43] focus on identifying optimal positions for language-vision alignment. Additionally, some methods [34; 5] have enhanced alignment between language and pixel features by designing specialized loss functions. Other approaches [21; 45] treat referring image

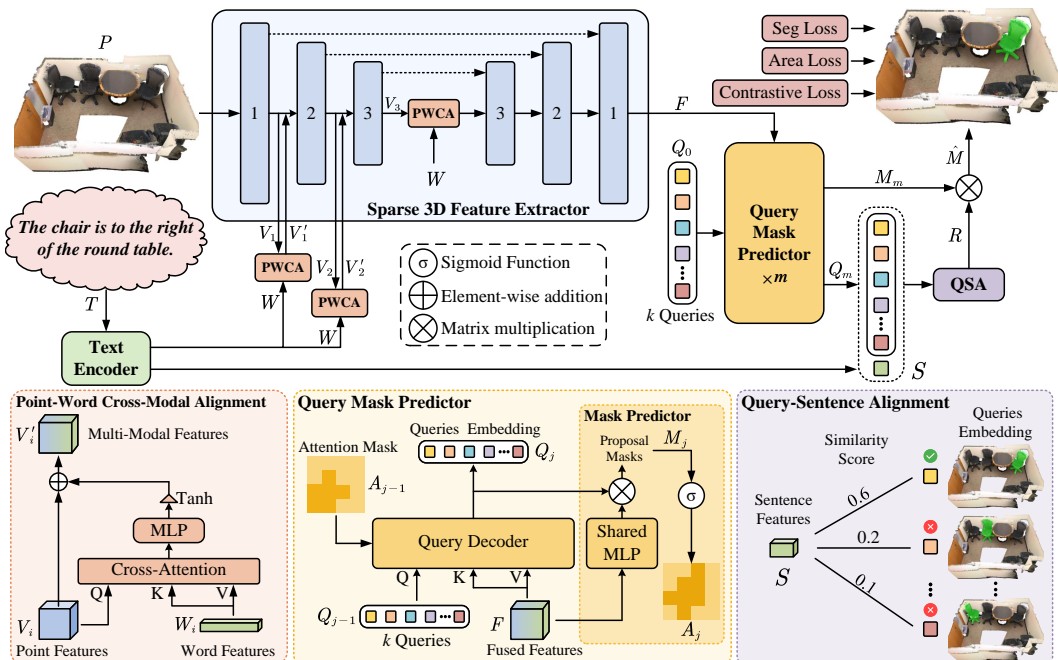

Figure 2: Overview of our **LESS** framework. Given a point cloud scene $P$, we use a sparse 3D feature extractor to extract multi-scale feature $V_i$. The query $T$ is sent to a text encoder and we obtain the word features $W$ and sentence features $S$. Meanwhile, we introduce a PWCA module aligns the word features $W$ with the multi-scale point cloud features $V_i$. After that, an $m$-layer QMP module is adopted to decode $K$ learnable queries $Q_0$ base on the fused feature $F$, and output query embeddings $Q_m$ and proposal masks $M_m$. Finally, QSA module aligns the query embeddings $Q_m$ with sentence features $S$, i.e., computes the similarity scores $R$ that filter the proposal masks $M_m$ to the final mask prediction $\hat{M}$.

segmentation as an auto-regressive vector generation problem, creating masks from generated closed vectors.

Methods for referring image segmentation have been extensively explored, yet they cannot be directly applied to Referring 3D Segmentation due to the inherent challenges of point cloud scenes. Additionally, unlike referring image segmentation methods that typically employ pre-trained visual encoders, Referring 3D Segmentation lacks such resources, compelling reliance on limited supervisory signals for model training.

## 3 Method

The overall framework of our proposed LESS is shown in Fig.2, which leverages Point-Word Cross-Modal Alignment and Query-Sentence Alignment to facilitate multi-modal interaction. In this section, we start by introducing our visual and text feature extractor in Sec.3.1. Then Query Mask Predictor and Query-Sentence Alignment is detailed in Sec.3.2 and Sec.3.3. Finally in Sec.3.4, we introduce our area regularization loss and point-to-point contrastive loss.

### 3.1 Visual and Text Feature Extractor

**Sparse 3D Feature Extractor.** The point cloud scene $P \in \mathbb{R}^{N \times 6}$ contains $N$ points in the scene, and each point is represented with six dimensions of RGBXYZ attributes. We first voxelize points into regular voxels and adopt a sparse 3D U-Net [31; 15] to extract point-wise fused feature $F \in \mathbb{R}^{N \times C}$. Here the encoder part of the U-Net has 5 stages and the feature from the $i$-th encoder stage is denoted as $V_i$.

**Text Encoder.** Given the query sentence $T$ with $L$ words, a text encoder is used to embed the query into $C$-dimensional feature vectors. In this paper we choose GRU [6], BETR [8] and RoBERTa [22] as our text encoder respectively and fine-tune the BERT or RoBERTa during training. Finally, we can get both word features $W \in \mathbb{R}^{L \times C}$ and sentence features $S \in \mathbb{R}^C$ after the text encoder.

**Point-Word Cross-Modal Alignment.** Due to the lack of such rich annotations as [13; 29], it is crucial for our model to learn the relationship between fine-grained word-level features and point-level features in such a point-level segmentation task. Meanwhile, we notice that leveraging the rich convolutional layers in the encoder to excavate multi-modal context is effective way to extract language-aware visual feature. Therefore, we design Point-Word Cross-Modal Alignment (PWCA) module, as shown in the lower left part in Fig.2, which contains a standard cross-attention module [32] and an MLP-Tanh gate. Cross-attention module aligns point-wise and word-wise feature to get the language-aware visual feature. A nonlinear tanh gate is adopted to prevent the fused signal from overwhelming the original signal. Given the multi-scale point cloud features $\{V_i \in \mathbb{R}^{N_i \times C_i}\}_{i=1}^5$, we simply project the word feature $W \in \mathbb{R}^{L \times C}$ to $W_i \in \mathbb{R}^{L \times C_i}$. PWCA can be formulated as follow:

$$V_i' = \text{Tanh}(\text{MLP}(\text{CrossAttn}(V_i, W_i)) + V_i, \quad i \in \{1, ..., 5\}, \tag{1}$$

where $i$ indicates the $i$-th stage of the encoder part of our sparse 3D feature extractor. Here we use $V_i$ as the query and $W_i$ as the key and value for cross attention.

## 3.2 Query Mask Predictor

Inspired by [31; 30], Query Mask Predictor (QMP) module, as shown in Fig.2, takes fused feature $F \in \mathbb{R}^{N \times C}$ and learnable queries $Q_0 \in \mathbb{R}^{K \times C}$ as input and progressively distinguishes the referring target by multi-layer cross-modal transformers. Finally, we extract the proposal masks $M_m \in \mathbb{R}^{K \times N}$ based on queries embeddings $Q_m \in \mathbb{R}^{K \times C}$ and fused feature $F$.

**Query Decoder.** As illustrated in the lower middle section of Fig.2. Here query decoder is comprised $m$-layer masked cross-attention modules [32], where the fused features $F$ are served as keys and values and $A_{j-1} \in \{0,1\}^{K \times N}$ is utilized as the attention mask. Therefore, in the $j$-th layer, the learnable queries $Q_{j-1}$ capture multi-modal contextual information from fused feature $F$ via a query decoder to obtain the query embeddings $Q_j \in \mathbb{R}^{K \times C}$.

**Mask Predictor.** First the fused feature $F \in \mathbb{R}^{N \times C}$ is processed by a Shared MLP, which indicates each Query Mask Predictor layer shares the same MLP. We perform the matrix multiplication on the new $F$ and $Q_j \in \mathbb{R}^{K \times C}$ to generate proposal mask predictions $M_j \in \mathbb{R}^{K \times N}$. After applying a sigmoid function whose threshold of 0.5, we can get the new binary attention mask $A_j \in \{0,1\}^{K \times N}$.

Finally, the Query Mask Predictor is formally described as follows:

$$Q_j = \text{Query Decoder}_j(Q_{j-1}, A_{j-1}, F), \tag{2}$$

$$M_j = Q_j \otimes \text{Shared-MLP}(F)^\top, \tag{3}$$

$$A_j = \begin{cases} 1 & \sigma(M_j) \geq 0.5 \\ 0 & \text{otherwise} \end{cases}, \tag{4}$$

where $\top$ denotes the transpose operation, $\otimes$ represents matrix multiplication, and $\sigma$ refers to the sigmoid function. Here we need to note that the $Q_0$ is randomly initialized. Therefore we can leverage the function (3-4) to initialize the attention mask $A_0$.

## 3.3 Query-Sentence Alignment

The Referring 3D Segmentation task involves the segmentation of a solitary target object according to the query. Previous modules focus on extracting language-aware visual feature from aligning point-wise feature and word-wise feature, which lacks a comprehensive perception of the entire query sentence. Therefore, we adopt Query-Sentence Alignment (QSA) to better align the query feature with sentence-level feature. We perform the matrix multiplication on $Q_m \in \mathbb{R}^{K \times C}$ and $S \in \mathbb{R}^C$ to get the their similarity score $R \in [0,1]^K$. Formally, Query-Sentence Alignment can be represented as:

$$R = \text{Softmax}(Q_m \otimes S). \tag{5}$$

The final mask prediction $\hat{M} \in \mathbb{R}^N$ is produced by weighted sum of similarity score $R$ and proposed mask prediction $M_m \in \mathbb{R}^{K \times N}$. Finally, we use a sigmoid function and a threshold of $0.5$ to produce the final predicted binary mask $\hat{Y} \in \{0,1\}^N$ :

$$\hat{M} = R \otimes M_m, \qquad \hat{Y} = \begin{cases} 1 & \sigma(\hat{M}) \geq 0.5 \\ 0 & \text{otherwise} \end{cases}. \tag{6}$$

### 3.4 Loss Function

**Segmentation Loss.** Different from previous work [13; 29], we take the Referring 3D Segmentation task as segmentation task with only binary mask $Y \in \{0,1\}^N$. Here we utilize the Binary Cross-Entropy (BCE) loss function to compute the segmentation loss, which can be formulated as:

$$\mathcal{L}_{\text{seg}} = \text{BCE}(\sigma(\hat{M}), Y). \tag{7}$$

**Area Regularization Loss.** For Referring 3D Segmentation task, each query always corresponds to one target object in the point cloud scene. The target objects occupy a smaller area in the large scale of 3D point cloud. As a result, the predicted mask often includes backgrounds or other objects. To address this, we propose a region regularization loss, which promotes the network to predict a smallest mask by minimize the output probability of each point, formulated as:

$$\mathcal{L}_{\text{area}} = \frac{1}{N} \sum_{i=1}^{N} \sigma(\hat{M}_i). \tag{8}$$

By combining with the segmentation loss, we intend to segment only the most probable regions while reducing segmentation of large-scale irrelevant background areas.

**Point-to-Point Contrastive Loss.** Area regularization loss uniformly penalizes the predicted probabilities of all points, which can reduce the majority of the background points. However, the network struggles to differentiate between objects that possess characteristics similar to those described target object in the latent space. Therefore, we propose a point-to-point contrastive loss [26] that pull the points from the described object together and push away the rest points:

$$\mathcal{L}_{\text{p2p}} = -\frac{1}{|\mathcal{P}|} \sum_{i=1}^{|\mathcal{P}|} \frac{\exp(\mathbf{P}_i \cdot \mathbf{P}_{\text{avg}}/\tau)}{\exp(\mathbf{P}_i \cdot \mathbf{P}_{\text{avg}}/\tau) + \sum_{j=1}^{|\mathcal{N}|} \exp(\mathbf{P}_i \cdot \mathbf{N}_j/\tau)}, \tag{9}$$

where $\mathcal{P}$ is the positive point set from the described object, $\mathcal{N}$ is the negative point set from the background, $\mathbf{P}_i$ denotes the L2-normalized feature vector of $i$ -th positive points from $F$, while $\mathbf{N}_j$ denotes the L2-normalized feature vector of $j$-th negative points from $F$, $\mathbf{P}_{\text{avg}}$ is the average feature vector of positive point $\mathbf{P}_{\text{avg}} = \frac{1}{\mathcal{P}} \sum_{i=1}^{|\mathcal{P}|} \mathbf{P}_i$, and $\tau$ is the hyper-parameter. The contrastive loss promotes the network to distinguish the described object from the adjacent background points in a fined-grained manner.

The overall loss function is the weighted sum of the aforementioned three loss functions:

$$\mathcal{L} = \lambda_{\text{seg}} \mathcal{L}_{\text{seg}} + \lambda_{\text{area}} \mathcal{L}_{\text{area}} + \lambda_{\text{p2p}} \mathcal{L}_{\text{p2p}}, \tag{10}$$

where $\{\lambda_{\text{seg}}, \lambda_{\text{area}}, \lambda_{\text{p2p}}\}$ is set to $\{1, 1, 0.05\}$ in practice to balance the contrastive loss because of the large amount of points.

## 4 Experiments

### 4.1 Dataset and Experiment Settings

**ScanRefer.** ScanRefer [3] is a dataset for 3D referring expression comprehension tasks such as 3D visual grounding and 3D referring instance segmentation. It contains $51,583$ queries of $11,046$ objects from $800$ ScanNet [7] scenes. Each scene contains $13.81$ objects and $64.48$ queries on average.

Table 1: Quantitative results of different methods on ScanRefer [3] validation set. "Supervision" indicates the type of supervision. **Ins.** denotes instance labels and **Sem.** indicates semantic labels. **Mask** represents binary labels. **Bold** indicates the best.

|  | Method | Backbone | Label Effort‡ | Supervision | mIoU | Acc@0.25 | Acc@0.5 |
|---|---|---|---|---|---|---|---|
| Two Stage | TGNN | GRU |  | Ins.+ Sem. | 26.10 | 35.00 | 29.00 |
|  | TGNN | BERT | > 20 min | Ins.+ Sem. | 27.80 | 37.50 | 31.40 |
|  | X-RefSeg | GRU |  | Ins.+ Sem. | 29.77 | 39.85 | 33.52 |
|  | X-RefSeg | BERT |  | Ins.+ Sem. | 29.94 | 40.33 | 33.77 |
| Single Stage | LESS (ours) | GRU |  | Mask | 32.19 | 51.00 | 26.41 |
|  | LESS (ours) | BERT | < 2 min | Mask | 32.44 | 51.41 | 29.02 |
|  | LESS (ours) | RoBERTa |  | Mask | **33.74** | **53.23** | **29.88** |

‡ The evaluate of label effort is base on a single sample.

Table 2: Module ablation on ScanRefer dataset.

|  | PWCA | QSA | mIoU | A@25 | A@50 |
|---|---|---|---|---|---|
| (a) |  |  | 32.66 | 51.71 | 27.20 |
| (b) | ✓ |  | 33.44 | 52.73 | 28.92 |
| (c) | ✓ | ✓ | **33.74** | **53.23** | **29.88** |

Table 3: Loss ablation on ScanRefer dataset.

|  | $\mathcal{L}_{area}$ | $\mathcal{L}_{p2p}$ | mIoU | A@25 | A@50 |
|---|---|---|---|---|---|
| (a) |  |  | 25.86 | 40.85 | 16.81 |
| (b) | ✓ |  | 31.04 | 49.61 | 24.72 |
| (c) | ✓ | ✓ | **33.74** | **53.23** | **29.88** |

**Evaluation Metric.** Following previous work [13; 29], we use the mean intersection-over-union (**mIoU**), and **Acc@kIOU** as the evaluation metrics. The mIoU is the average of the IoU over all test samples and the Acc@kIOU measures the accuracy of test samples with an IoU higher than the threshold $k$, where $k \in \{0.25, 0.5\}$. We use A@25 and A@50 for brevity in some of the following tables.

**Implementation Details.** We adopt the 3D spares U-Net [31] as our 3D feature extractor. we explore multiple text encoders, i.e., GRU [6], BERT [8] and RoBERTa [22] in our experiments for comparative analysis. For BERT and RoBERTa, we use the official pre-trained weights and fine-tune them during training. We set the number of queries $K$ to 20 and use a single layer for QMP module. We set an initial learning rate of 2e-5 for the text encoder and 1e-4 for the others. We reduce the learning rate by a multiplicative factor of 0.95 each epoch and adopt Adam [24] as our optimizer. The weights of our loss function $\{\lambda_{\text{seg}}, \lambda_{\text{area}}, \lambda_{\text{p2p}}\}$ is set to $\{1, 1, 0.05\}$. We train for 64 epochs with a batchsize of 14, and all experiments are implemented on PyTorch [27].

## 4.2 Comparison with State-of-the-Art Methods

As shown in Tab.1, we evaluate our LESS against the previous two-stage methods. LESS outperforms the previous SOTA method using GRU [6] and BERT [8] by an impressive progress of 2.42%, and 2.50% on mIoU, and 11.15% and 11.08% on Acc@0.25 respectively. Moreover, we conduct an extra experiment using RoBERTa [22] , outperforming the best method of 3.8% and 12.9% on mIoU and Acc@0.25 respectively. Such results demonstrate the potential of our method. We also reported the comparison between our method and the two-stage approach in terms of label effort. We find that our label-efficient method saves more than 90% label effort compared to existing methods.

However we find that the performance of our LESS has a gap on Acc@0.5 compared to previous methods. Previous methods employed a segmentation-matching strategy. Once matching successfully, the IoU between predicted mask and ground truth is mostly higher than 0.5, which is beneficial for the Acc@0.5. In contrast, our single-stage method without instance labels and semantic labels can not extract the more accurate instance candidates as prior knowledge to assist referring 3D segmentation task. Therefore, it is acceptable for our method to perform lower than previous methods on Acc@0.5 with fewer supervisory signals.

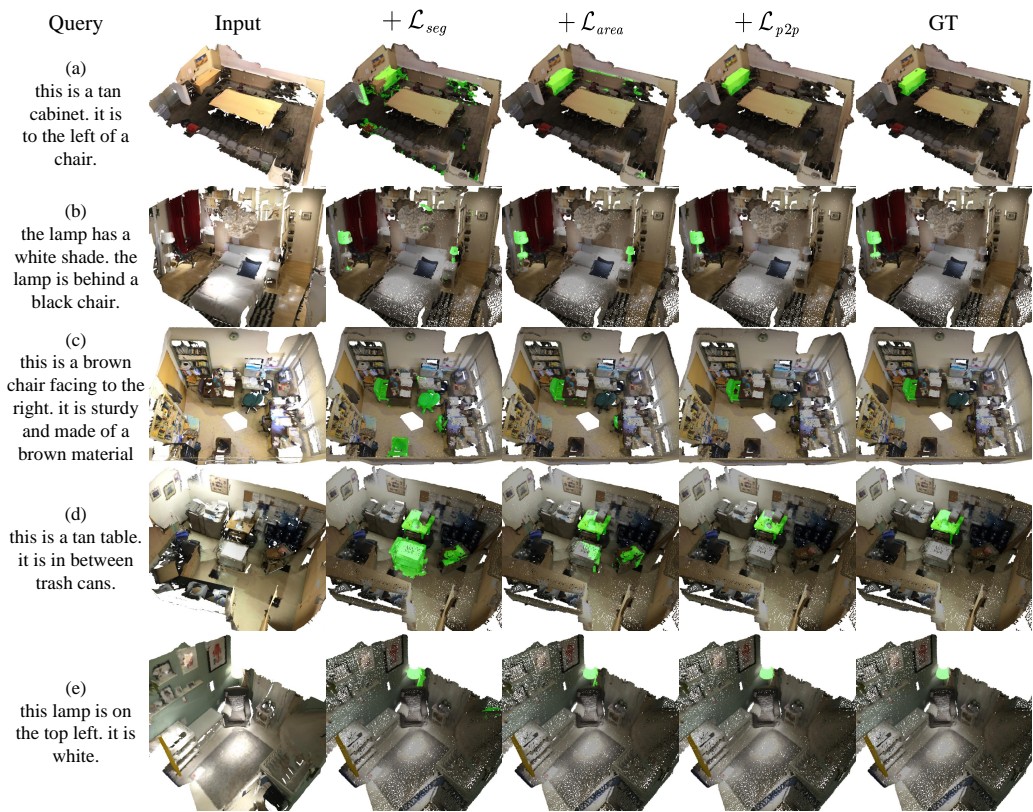

| Query | Input | $+\mathcal{L}_{seg}$ | $+\mathcal{L}_{area}$ | $+\mathcal{L}_{p2p}$ | GT |

(a) this is a tan cabinet. it is to the left of a chair.

(b) the lamp has a white shade. the lamp is behind a black chair.

(c) this is a brown chair facing to the right. it is sturdy and made of a brown material

(d) this is a tan table. it is in between trash cans.

(e) this lamp is on the top left. it is white.

Figure 3: Final predictions using different combinations of loss functions. The queries and input scenes are shown in column 1 and 2. Columns 3 to 5 indicate the gradual addition of loss functions.

## 4.3 Ablation Studies.

We conducted several ablation studies to evaluate the effectiveness of the key components in our proposed network. In these studies, all other components and hyper-parameter settings are kept consistent with the aforementioned experiments, except for the component being ablated.

**Module Ablation.** In this ablation study, we evaluate the effectiveness of the QSA and PWCA modules. As indicated in Table2, the ablation model (a) only retain the sparse 3D feature extractor and query mask predictor. Here we perform element-wise addition of the word features $W$ to the multi-scale features $\{V_i\}_{i=1}^{5}$ instead of PWCA and project the proposed mask prediction $M_m \in \mathbb{R}^{K \times N}$ to the final mask prediction $\hat{M} \in \mathbb{R}^N$ via an MLP instead of QSA. We set model (a) as the baseline of our experiment. Compared to model (a), model (b) adopts PWCA module and we find that the model performance increase greatly from 32.66% to 33.44%. This observation proves that PWCA facilitate fine-grained cross-modal alignment between points and words, which is more effectively to leverage the rich convolutional layers in the encoder to excavate multi-modal context. When we introduce QSA to model (b), as shown in model (c), we can find that the performance of mIoU is improved from 33.44% to 33.74%. Such results indicates that the fine-grained point-word alignment of PWCA and the coarse-grained query-sentence alignment of QSA effectively coupled to enhance the capability in capturing multi-modal context.

**Loss Ablation.** As demonstrated in 3.4, we refine predicted masks from coarse to fine by introducing two loss functions, i.e., area regularization loss $\mathcal{L}_{area}$ and point-to-point contrastive loss $\mathcal{L}_{p2p}$. We successively add the loss functions and the results are shown in Tab.3. The model (a) is only supervised by segmentation loss $\mathcal{L}_{seg}$. When we introduce the $\mathcal{L}_{area}$ into model (a), as shown in model (b), the performance greatly increase from 25.86% to 31.04% on mIoU, which indicates our

Table 4: The impact of linguistic features at different granularities. **Word** represent word-level features, and **Sentence** represent sentence-level features.

| Word | Sentence | mIoU | Acc@0.25 | Acc@0.5 |
|:---:|:---:|:---:|:---:|:---:|
| ✓ | | 31.56 | 49.50 | 25.84 |
| | ✓ | 33.02 | 52.33 | 27.75 |
| ✓ | ✓ | **33.74** | **53.23** | **29.88** |

Table 5: The impact of the number of queries.

| Num of queries | mIoU | A@25 | A@50 |
|:---:|:---:|:---:|:---:|
| 20 | **33.74** | **53.23** | **29.88** |
| 60 | 32.62 | 51.96 | 27.70 |
| 100 | 33.16 | 52.98 | 28.82 |

Table 6: The impact of the number of layers.

| Num of layers | mIoU | A@25 | A@50 |
|:---:|:---:|:---:|:---:|
| 1 | **33.74** | **53.23** | **29.88** |
| 3 | 32.45 | 51.09 | 27.70 |
| 6 | 32.66 | 52.18 | 28.33 |

area loss can exclude a significant number of irrelevant background points. Moreover, we successively add the contrastive loss $\mathcal{L}_{p2p}$, as shown in the model (c). The performance is improved from 31.04% to 33.74%, which proves that the contrastive loss can make the model more focused on the target area rather than others.

Qualitative results are shown in Fig.3, it indicates that: **i)** When only the segmentation loss $\mathcal{L}_{seg}$ is applied *(column 3)*, the predicted masks include many points from other regions. **ii)** After adding the $\mathcal{L}_{area}$, most of the irrelevant points disappear *(row a, row d)*. **iii)** After incorporating the $\mathcal{L}_{p2p}$, objects that were previously difficult to distinguish due to their similarity are successfully separated, and the predictions are close to the ground-truth.

Both quantitative and qualitative experiment demonstrate that our proposed loss function effectively reduces large-scale background misclassifications and distinguish objects or points with the similar characteristics.

### 4.4 Extension Experiments

**Linguistic Features at Different Levels of Granularity.** In this section, we will investigate the impact of linguistic features at different levels of granularity, as shown in Table.4. The first row represent that we leverage word-level features in both PWCA module and QSA module as text features. The second row indicates we utilize sentence-level features in both modules. We can find that sentence-only method even outperforms the word-only one. Referring 3D Segmentation task involves the segmentation of a solitary target object. This mandates a more profound and thorough comprehension of the semantic information conveyed by the sentence, extending beyond a mere focus on its individual words. The last row indicates the word-level features are utilized in PWCA module and sentence-level features are leveraged in QSA module. We can find that the performance of it outperforms two introduced above, which proves that fine-grained word-level feature is also helpful in extracting the 3D language-aware visual feature.

**Layer and Query Number of QMP.** We also investigate the performance on different query and layer numbers in the QMP module. As shown in Tab.5 and Tab.6, we find that too many queries and layers do not bring performance gains for Referring 3D Segmentation task. As a result, we set 20 queries and 1 QMP layer as the default configuration after balancing performance and efficiency.

### 4.5 Limitations

The limitations of LESS due to the inherent complexity of 3D point clouds and the ambiguous queries, although we have made significant improvements on previous methods. The scarcity of detailed semantic annotations are still challenging our model from distinguishing multiple similar objects. These limitations could guide our future work.

## 5 Conclusions

In this paper, we propose LESS, a label-efficient and single-stage approach for Referring 3D Segmentation. Specifically, our LESS enhances feature extraction by integrating multi-modal features and employs progressive constraints on predicted masks, achieving fine-grained alignment between points and words and distinguishing between points or objects with similar characteristics. Comprehensive experiments demonstrate that our single-stage method outperforms existing two-stage approaches on ScanRefer dataset, using only the binary labels as supervision. Though our framework still has some limitations, we believe that solving the Referring 3D Segmentation task only using the binary labels is a new and promising research, and we hope that LESS can serve as a simple but powerful baseline to inspire future research on Referring 3D Segmentation.

## 6 Acknowledgements

This work was supported in part by the National Natural Science Foundation of China under Grants 62371310, in part by the Guangdong Basic and Applied Basic Research Foundation under Grant 2023A1515011236, in part by the Stable Support Project of Shenzhen (Project No.20231122122722001).

We also thank the support by the MUR PNRR project FAIR (PE00000013) funded by the NextGenerationEU, the PRIN project CREATIVE (Prot. 2020ZSL9F9), and the EU Horizon project ELIAS (No. 101120237). We also acknowledge the CINECA award under the ISCRA initiative, for the availability of partial HPC resources support.

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

# Appendix

## A   Semantic and Instance Labels vs. Binary Labels

The differences between semantic labels, instance labels, and binary labels are shown in Fig.4.

Semantic labels assign a category ID to each point in a scene, with points belonging to the same category (e.g., chairs) sharing the same ID. Instance labels assign a object ID to each point within the same object (e.g., a specific chair), distinguishing different objects by assigning different instance IDs. Binary labels assign a Boolean value to each point in the scene; points that are part of the described object are assigned a value of 1, and points that are not are assigned a value of 0.

Therefore, compared to binary labels, the annotations of semantic labels and instance labels are more time consuming and labor intensive.

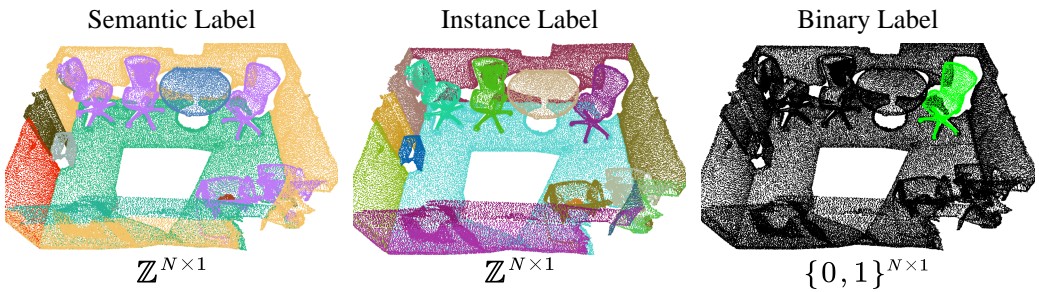

Figure 4: There different types of labels.

## B   More Qualitative Results

We present our success cases and failure cases in Fig.5. Our method accurately segments objects with clear queries. However, ambiguous descriptions can still confuse our model and leads to segment both the referred object and other similar objects.

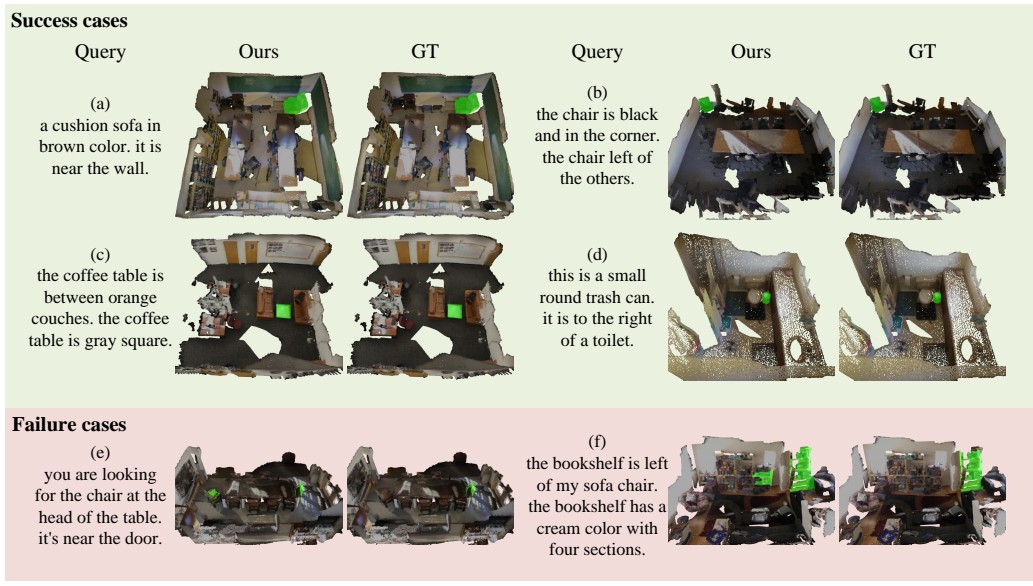

Figure 5: Qualitative results of both the success cases and failure cases of our LESS.

## C   Time Consumption Comparison

As shown in Tab.7, we evaluate the training and inference time both of TGNN[13] and X-RefSeg[29]. All experiments are conducted on an NVIDIA 4090 GPU and the number of batch sizes and epoch of three methods are kept the same. For the two-stage training and inference of TGNN and X-RefSeg, we followed the settings in their open source codes. We can find that our LESS consumes less time than both of TGNN and X-RefSeg in both training and inference.

Table 7: The comparison of inference time and training time with previous work.

| Method | Inference (Whole Dataset) (min) | Inference (Per Scan) (ms) | Training (Stage 1) (h) | Training (Stage 2) (h) | Training (All) (h) |
|---|---|---|---|---|---|
| TGNN | 27.98 | 176.57 | 156.02 | 8.53 | 164.55 |
| X-RefSeg | 20.00 | 126.23 | 156.02 | 7.59 | 163.61 |
| Ours | **7.09** | **44.76** | - | - | **40.89** |

## D   More Quantitative Results

A concurrent work 3D-STMN [35] utilizes a pre-trained 3D feature extractor [31] to perform referring 3D segmentation. Given that their backbone is pre-trained on an instance segmentation task with semantic and instance label, it is reasonable to conclude that their approach cannot be considered a label-efficient and single-stage method. For fair comparison, we follow the settings in their open source code and train their network from scratch, except for BERT [8] module. As shown in Tab.8, our method overtakes 3D-STMN by 11.34%, 18.27% and 12.65% on mIoU, Acc@0.25 and Acc@0.5 respectively.

Table 8: Quantitative results of 3D-STMN [35] and ours.

| Method | mIoU | Acc@0.25 | Acc@0.5 |
|---|---|---|---|
| 3D-STMN (from scratch) | 22.40 | 34.96 | 17.23 |
| Ours | **33.74** | **53.23** | **29.88** |

## E   More Ablation Results

### E.1   Query Number

We further conduct more ablation studies in terms of the number of queries on ScanRefer. As shown in Tab.9, we ablate the number of queries as 5, 15, 20, it can be observed that keeping 20 queries brings higher accuracy while lower accuracy when using fewer queries. We suppose that fewer queries can not help to learn comprehensive feature patterns while an appropriate number of queries is enough to cover the needed feature patterns.

Table 9: The impact of the number of queries.

| Num of queries | mIoU | Acc@0.25 | Acc@0.5 |
|---|---|---|---|
| 5 | 32.75 | 51.81 | 28.68 |
| 15 | 33.27 | 52.56 | 28.97 |
| 20 | **33.74** | **53.23** | **29.88** |

### E.2   Mask Selection Strategy

We also conduct the ablation study which selects the mask with the highest score during QSA process as shown in Tab.10. The results demonstrate that the aggregation of multiple masks yields superior

performance compared to the selection of a single, highest-ranked mask. This approach facilitates the capture of subtle nuances and intricate details that may be overlooked when relying on a single mask alone. The incorporation of multiple masks offers a more comprehensive and precise representation, ultimately enhancing the accuracy of the final model prediction.

Table 10: The impact of mask selection strategy.

| Num of queries | mIoU | Acc@0.25 | Acc@0.5 |
|---|---|---|---|
| Top-1 | 33.18 | 52.93 | 28.65 |
| Ours | **33.74** | **53.23** | **29.88** |

## F  Potential Negative Societal Impacts

Our method has no ethical risk on datasets usage and privacy violation because all the datasets and tools are publicly available and transparent.

