# OpenReview forum: "LESS: Label-Efficient and Single-Stage Referring 3D Segmentation"
_NeurIPS.cc/2024/Conference — NeurIPS 2024 poster_

### Official Review · Reviewer_Monv · 2024-07-04

**Soundness:** 3
**Presentation:** 3
**Contribution:** 3
**Rating:** 7
**Confidence:** 5

**Summary:**

In this paper, the authors propose a Label-Efficient and Single-Stage referring 3D instance segmentation method, which is under the supervision of binary mask. They propose Point-Word Cross-Modal Alignment, Query Mask Predictor and Query Alignment modules to achieve cross-modal alignment. Besides, the area regularization loss and the point-to-point contrastive loss are introduced to eliminate interference caused by multiple objects and backgrounds. Experimental results show that the proposed method outperforms the existing state-of-the-art method on ScanRefer dataset with only the supervision of binary. This is a nice paper dealing an interesting research problem, presenting insights, and being well-written.

**Strengths:**

1.	This paper firstly investigated the single stage referring 3D instance segmentation task to bridge the gap between detection and matching, only under the supervision of binary mask. The proposed method can provide valuable insights to the further research in multi-modal 3D scene understanding.
2.	To align features among different modalities, such as fined-grained alignment between point and word features, coarse-grained alignment between masks and sentences, several alignments modules are well designed, which exhibit a certain level of novelty.
3.	Experimental results show that the proposed method outperforms the existing state-of-the-art method on ScanRefer dataset with only the supervision of binary label.
4.	The paper is well-written and is easy to follow.

**Weaknesses:**

1.	Methods TGNN and X-RefSeg are not evaluated with the RoBERTa backbone.
2.	The explanation of why more queries and layers as shown in Tab. 5 and 6 do not bring performance gains are missing.

**Questions:**

1.	Why the proposed alignment modules work well?
2.	How about the training time and inference time comparison?

**Limitations:**

No negative societal impact

---

> ### Author Rebuttal · Authors · 2024-08-07
>
> We appreciate that you recognize the significance of our work. We will respond to your concerns in the following:
>
> ## Q1: For the validation of TGNN and X-RefSeg with the RoBERTa backbone
>
> Thanks for your detailed reading and pointing out this. We reimplement the TGNN and X-RefSeg with the same RoBERTa backbone as ours to conduct experiments on ScanRefer benchmark, since these exisitng works have not applied RoBERTa backbone as language module for their experiments. We then follow their settings to train and validate the performances. As shown in the table below, we can obtain their newly corresponding results, 28.83% mIoU for TGNN (RoBERTa) and 30.72% mIoU for X-RefSeg (RoBERTa), which still shows inferior performance to ours 33.74%, which further validates the superiority of our single-stage prardigm. Besides, we observe that when the loss coefficient is set to 10, our LESS outperforms the performance of TGNN and X-RefSeg on all the metrics. More discussion will be added in our paper.
>
> |               Method               |   mIoU    |  Acc@25   |  Acc@50   |
> | :--------------------------------: | :-------: | :-------: | :-------: |
> |             TGNN (GRU)             |   26.10   |   35.00   |   29.00   |
> |            TGNN (BERT)             |   27.80   |   37.50   |   31.40   |
> |           TGNN (RoBERTa)           |   28.83   |   39.15   |   32.50   |
> |           X-RefSeg (GRU)           |   29.77   |   39.85   |   33.52   |
> |          X-RefSeg (BERT)           |   29.94   |   40.33   |   33.77   |
> |         X-RefSeg (RoBERTa)         |   30.72   |   41.54   |   34.42   |
> | Ours (RoBERTa) $\lambda_{area}=1$  |   33.74   | **53.23** |   29.88   |
> | Ours (RoBERTa) $\lambda_{area}=10$ | **35.08** |   52.30   | **34.43** |
>
> ## Q2: For the more querie and layers do not bring performance gains
>
> Thanks for pointing out this. With an excessive number of queries, too many queries become redundant, leading to overlapping or duplicate proposal masks. This redundancy does not contribute to better segmentation and can instead introduce noise, even degrading performance improvements. Besides, Increasing the number of layers can lead to overfitting, especially with the limited supervisory signals available in our label-efficient setting, binay mask supervision.
>
> ## Q3: For reasons for good performance of proposed aligment modules
>
> Thanks for your detailed reading and pointing out this. Point-Word Cross-Modal Alignment module aligns fined-grained point and word features. Also Query Mask Predictor module and Sentence Query Alignment modules make the coarse-grained alignment between masks and sentences. The coarse-grained  sentences-masks and fine-grained words-points effectively coupled to enhance the capability in capturing multi-modal context, which is helpful for our aligment modules.
>
> ## Q4: For the training time and inference time
>
> Thanks for your detailed reading and pointing out this.  As shown at General Respons and in the following table, we evaluate the training and inference time both of TGNN and X-RefSeg.  All experiments are conducted on an NVIDIA 4090 GPU and the batch sizes of three methods are kept the same. For the two-stage training and inference of TGNN and X-RefSeg, we followed the settings in their open source codes. We can find that our LESS consumes less time than both of TGNN and X-RefSeg in both training and inference.
>
> |  Method  | Inference (Whole Dataset) (min) | Inference (Per Scan) (ms) | Training (Stage 1) (h) | Training (Stage 2) (h) | Training (All) (h) |
> | :------: | :-----------------------------: | :-----------------------: | :--------------------: | :--------------------: | :----------------: |
> |   TGNN   |              27.98              |          176.57           |         156.02         |          8.53          |       164.55       |
> | X-RefSeg |              20.00              |          126.23           |         156.02         |          7.59          |       163.61       |
> |   Ours   |            **7.09**             |         **44.76**         |           -            |           -            |     **40.89**      |
>
> Thanks again for these constructive comments, we will take all these experimental results or further discussions into consideration into our revised version. If any questions, please kindly let us know.

---

> > ### Comment · Reviewer_Monv · 2024-08-14
> >
> > Thank the authors for the rebuttal.
> >
> > First, my comments were clearly addressed. Specifically, the validation of TGNN and X-RefSeg with the RoBERTa backbone, and the comparisons of the training time and inference time demonstrate the proposed method achieves the best performance. Besides, the explanations on the performance gains and motivation of the proposed alignment module makes the contributions of the proposed method clearer.
> >
> > Second, I read the other reviewers’ comments as well as the rebuttal information. For the raised questions, such as the validation on other dataset, ablation studies on the mask selection strategy, explanation of the significant differences between the proposed modules and existing techniques and concerns to label efficiency, the authors also responded accordingly.
> >
> > In summary, this work is relatively novel, which investigated the single stage referring 3D instance segmentation task to bridge the gap between detection and matching, only under the supervision of binary mask. The proposed method can provide valuable insights to the further research in multi-modal 3D scene understanding. Therefore, I am maintaining my original score.

---

> > > ### Author Response · Authors · 2024-08-14
> > > **Response to Reviewer Monv**
> > >
> > > We are grateful for your comprehensive and encouraging review and acknowledgement. We are pleased that you appreciate the technical contributions of our work, which could motivate us to explore more interesting works in the future.
> > >
> > > Thank you again for your support!

---

### Official Review · Reviewer_pLC3 · 2024-07-07

**Soundness:** 2
**Presentation:** 2
**Contribution:** 2
**Rating:** 5
**Confidence:** 3

**Summary:**

The paper proposes LESS, a single-stage, label-efficient pipeline for referring 3D instance segmentation. It introduces fine-grained and coarse-grained cross-modal alignment to improve feature matching and employs additional losses to reduce irrelevant predictions. Experiments are conducted on the ScanRefer dataset.

**Strengths:**

1. The motivation of this paper is meaningful. It proposes a label-efficient method for referring 3D instance segmentation, which can significantly reduce annotation costs.
2. The paper is well-written and easy to follow.
3. The experiments on the ScanRefer dataset show promising results.

**Weaknesses:**

1. The experiments are only conducted on the ScanRefer dataset, lacking validation on additional datasets like Nr3D/Sr3D used in TGNN[1].
2. In the Query-Sentence Alignment module, the final mask prediction is produced by a weighted sum of the similarity score and mask prediction. The rationale for using a weighted sum is not intuitive; why not select the mask with the highest score?
3. The paper lacks references to related literature. The Query Mask Predictor module is similar to the framework used in Mask3D[2], but Mask3D is not cited.

[1]. Text-guided graph neural networks for referring 3d instance segmentation, AAAI 2021.
[2]. Mask3d: Mask transformer for 3d semantic instance segmentation, ICRA 2023.

**Questions:**

See weaknesses

**Limitations:**

Yes

---

> ### Author Rebuttal · Authors · 2024-08-07
>
> Dear reviewer, thank you very much for your detailed and constructive comments.
>
> ## Q1: For the validation on other dataset
> Thanks for your detailed reading and pointing out this. We also conduct the experiments on the Nr3d and Sr3d datasets, as shown at General Response and in the following table. We can find that the performance of our LESS surpass the performance of TGNN by 3.6% and 1.8% on Nr3d and Sr3d respectively.  Such results present promising potential for the future works and shed new light on label-efficient and single-stage exploration for R3DIS task.
>
> |      | Method | Overall  |   Easy   |   Hard   | View-dep. | View-indep. |
> | :--: | :----: | :------: | :------: | :------: | :-------: | :---------: |
> | Nr3D |  TGNN  |   37.3   |   44.2   |   30.6   |   35.8    |    38.0     |
> |      |  Ours  | **40.9** | **47.4** | **35.2** | **39.7**  |  **41.9**   |
> | Sr3D |  TGNN  |   45.0   |   48.5   |   36.9   |   45.8    |    45.0     |
> |      |  Ours  | **46.8** | **50.5** | **37.8** | **46.6**  |  **46.3**   |
>
> ## Q2: For whether not select the mask with the highest score
> Thanks for your detailed reading and pointing out this.
>
> Firstly, using a weighted sum enables a soft decision-making process rather than a hard and binary choice. This can help in cases where multiple masks have similar scores, allowing the final prediction to be an aggregation of these maks, which can be more accurate than any single one.
>
> Besides, we also conduct the ablation study which selects the mask with the highest score in the ScanRefer dataset, as shown in the following table.  It can be shown that aggregating multiply masks leads to better performance than selecting only the highest one. This approach can help in capturing nuances and fine details that might be missed when relying on a solely single mask, which provide a more robust and accurate final prediction for the final model performance.
>
> |          |   mIoU    |  Acc@25   |  Acc@50   |
> | :------: | :-------: | :-------: | :-------: |
> |  Top-1   |   33.18   |   52.93   |   28.65   |
> | Baseline | **33.74** | **53.23** | **29.88** |
>
>
> ## Q3: For the cite of Mask3D
>
> Thanks for reminding us with this. We will add Mask3D[1] to our related literature and extend further discussions with Mask3D.
>
> [1]. Mask3d: Mask transformer for 3d semantic instance segmentation, ICRA 2023.
>
> Thanks again for these constructive comments, we will take all these experimental results or further discussions into consideration into our revised version. If any questions, please kindly let us know.

---

> > ### Comment · Reviewer_pLC3 · 2024-08-14
> >
> > Thanks for the authors' response. I have read all the contents, and most of my concerns have been addressed.
> > However, I have to admit that I'm not an expert in this field, so please consider more about the review opinions of reviewers with higher confidence.

---

> > > ### Author Response · Authors · 2024-08-14
> > > **Response to Reviewer pLC3**
> > >
> > > We are grateful for your comprehensive and encouraging review. Thank you.

---

### Official Review · Reviewer_6zjS · 2024-07-12

**Soundness:** 3
**Presentation:** 3
**Contribution:** 3
**Rating:** 5
**Confidence:** 5

**Summary:**

This paper proposes a label-efficient single-stage method for referring 3D instance segmentation. Specifically, this method enhances feature extraction by integrating multi-modal features, using only binary labels for supervision. It achieves fine-grained alignment between points and words, distinguishing points or objects with similar features.

**Strengths:**

This paper introduces the single-stage method pioneered by R3DIS, which I find very appealing.

The structure of the article is clear and the writing is fluent.

It achieves state-of-the-art (SOTA) performance on the ScanRefer dataset.

**Weaknesses:**

I agree with the authors' point that annotating precise instance-level labels in tasks like open-vocabulary or 3D visual grounding is time-consuming and labor-intensive. Therefore, it is impressive that this method reduces label effort by approximately 10 times while maintaining SOTA performance. However, the authors only used mIoU and Acc as metrics for semantic segmentation in the experiments, without using any 3D instance segmentation metrics (e.g., AP). I hope the authors can explain in detail why they did not use instance segmentation metrics. If no instance segmentation metrics were used, why is this task named 3D Instance Segmentation?

I hope the authors can validate their method's robustness across multiple datasets, as using only the ScanRefer dataset provides very limited persuasiveness.

**Questions:**

See Weaknesses part.

**Limitations:**

Yes.

---

> ### Author Rebuttal · Authors · 2024-08-07
>
> Thanks for your detailed and constructive comments. Next we address your questions.
>
> ## Q1: For the metrics of Referring 3D Instance Segmentation
>
> Thanks for pointing out this question. For the metrics of Referring 3D Instance Segmentation task, our experimental metrics are constantly followed by the previous works[1,2], called Acc@kIOU. Acc@kIOU measures the accuracy of test samples with an IoU higher than specific threshold k, where k ∈ {0.25, 0.5}. And we can find that the calculation method is much similar with the APx metric for common instance segmentation task. Hence, we choose the used Acc@kIOU in our main paper as the previous methods for the same Referring 3D Instance Segmentation task. We hope this explanation can help to clarify your confusion.
>
> [1]. Text-guided graph neural networks for referring 3d instance segmentation.
>
> [2]. X-RefSeg3D: Enhancing Referring 3D Instance Segmentation via Structured Cross-Modal Graph Neural Networks.
>
> ## Q2: For the validation on other dataset
>
> Thanks for your detailed reading and pointing out this. We also conduct the experiments on the Nr3d and Sr3d datasets, as shown at General Response and in the following table. We can find that the performance of our LESS surpass the performance of TGNN by 3.6% and 1.8% on Nr3d and Sr3d respectively.  Such results present promising potential for the future works and shed new light on label-efficient and single-stage exploration for R3DIS task.
>
> |      | Method | Overall  |   Easy   |   Hard   | View-dep. | View-indep. |
> | :--: | :----: | :------: | :------: | :------: | :-------: | :---------: |
> | Nr3D |  TGNN  |   37.3   |   44.2   |   30.6   |   35.8    |    38.0     |
> |      |  Ours  | **40.9** | **47.4** | **35.2** | **39.7**  |  **41.9**   |
> | Sr3D |  TGNN  |   45.0   |   48.5   |   36.9   |   45.8    |    45.0     |
> |      |  Ours  | **46.8** | **50.5** | **37.8** | **46.6**  |  **46.3**   |
>
> Thanks again for these constructive comments, we will take all these experimental results or further discussions into consideration into our revised version. If any questions, please kindly let us know.

---

> > ### Comment · Reviewer_6zjS · 2024-08-11
> > **Additional concerns to label efficiency**
> >
> > Thank you to the authors for addressing my questions one by one. However, I believe the first question was not adequately answered. While I understand that you have adopted the same metrics as in previous methods, I would like a clearer understanding of the deeper reason for not using the AP metric in instance segmentation. My guess is that it is because each prediction involves only one instance, so measuring accuracy based on IoU alone suffices.
> >
> > Additionally, there is an important issue I would like the authors to clarify. A core aspect of the paper is its emphasis on being label-efficient, and Table 1 compares the label effort of two-stage methods with that of the method proposed by the authors. To my knowledge, it takes approximately 25 minutes to semantically label an entire indoor scene on a per-instance basis, and about 2 minutes to label a single instance mask, which aligns with the time mentioned by the authors. However, training a typical instance segmentation network requires annotations for about 800 scenes from ScanNet, which totals approximately 25 * 800 = 20,000 minutes. On the other hand, as mentioned in the section "Dataset and Experiment Settings," training a rendering-based instance segmentation network requires 51,583 queries of 11,046 objects. Without considering the time for matching queries with objects, the total annotation time would be at least 2 * 11,046 = 22,092 minutes, which is more than the total time required for standard instance segmentation annotations. Therefore, based on this analysis, I find it unfair and unreasonable for the authors to conclude that their method is label-efficient by comparing only the annotation time for a single sample without considering the total number of samples required for training.
> >
> > I look forward to a more detailed explanation of the above questions. Thank you very much!

---

> > > ### Author Response · Authors · 2024-08-13
> > > **Response to Reviewer 6zjS**
> > >
> > > Thanks for your thoughtful reflections on this question and pointing out the confusion about the calculation and comparation of label efficiency. We will discuss these questions more deeply and below are our analysis and explanation:
> > >
> > > ## Q1: Rethinking whether not using the AP metric in instance segmentation
> > >
> > > - **Why the previous works[1,2] are called "instance segmentation" :** As we mentioned in the paper, the previous works adopt two-stage workaround. The first stage utilizes a 3D instance semantic segmentation network to get instance proposals. And the second stage leverages a network to match the query with corresponding instance proposals. Because the first stage is of great importance to this task, previous works simply named this task with  "instance segmentation", while we followed the previous works to do that and will extand more discussions into our revised version to rethink how to strictly name this task.
> > >
> > > - **Why not use AP metric in the R3DIS task:**  Thanks for this detailed reading and question. The target of referring 3D instance segmentation about a corresponding query is only one. Therefore the IoU metric is sufficient to measure this task. In addition, the Acc@x metrics also is used to measure the accuracy of the segmentation mask.  We will consider to add more comparative definition between Acc@x and AP into our revised implementation details.
> > >
> > > - **Rethinking about our LESS:**  We will take necessary in-depth discusions to explain and compare referring 3D instance segmentation with conventional instance segmentation task. Since our method is one-stage and lacks a first instance proposal stage, the title with "instance segmentation" may be not strictly align with the common instance segmentation task. However, the core contribution of our paper is to develop a simple yet effective single-stage for this community, shifting from complicated two-stage pipeline into single-stage one, which paves more spaces for future works. We will add more in-depth discussions to the final version of the paper and consider whether to revise the confusing title.
> > >
> > > ## Q2: For the labelling time on the whole dataset
> > >
> > > - **The number of scenes needed to annotate:**  Previous works [1,2] contain a 3D instance segmentation and a matching network. Based on their open source codes, we find that training a typical instance segmentation network requires annotations for **1513** scenes (train+val) rather than **800** scenes (which is the number of ScanRefer). Therefore, in light of the aforementioned assumption that the time required for the annotation of a single scene is 25 minutes, the overall time required for the training process should be 25 * 1,513 = 37,825 minutes, rather than the 20,000 minutes as estimated before.
> > >
> > > - **The number of objects needed to annotate:** In order to make a fairer and more reasonable comparison of label-efficiency between our LESS and previous works, we compare the number of objects required labelling. Firstly, we follow the open source codes of the previous works to process the dataset. From the processed dataset we find that the number of annotated objects under 1513 scenes is about **45,711**, even though excluding walls and floors. As for our LESS, we only need to annotate **11,046** objects, which is 4 times fewer compared to previous works. Here we need to note that the objects referred by ScanRefer is the subset of the objects in ScanNetv2.
> > >
> > > - **For the comparative method of time in the paper:** As we mentioned above, the objects referred by ScanRefer are a subset of all objects among all scenes. In the context of the time comparison presented in the paper, the comparison of labeling effort is based on a single sample.  This is because that, in consideration of reality scenarios,  it is often the case that the model is adapted to incorporate new data in order to align with evolving referential requirements.  Therefore, in our method, there only necessitate just a single or a few objects needed to be labelled, rather than the entire scene. When scaling up the dataset scale for this task, our advantages will become much more significant. As a result, we compared the label effort of a single sample in the paper. Thank you for pointing out this issue, and we will make further discussion on this in our paper to reduce any potential ambiguity.
> > >
> > > In summary, our method takes less time than the previous works in different comparison methods, which indicates our method is more label-efficient. Also We would like to extend our gratitude once more for your insightful suggestion. It is our sincere hope that the responses above could address your concerns. If you have any additional questions or suggestions, please let us know.
> > >
> > > [1]. Text-guided graph neural networks for referring 3d instance segmentation, AAAI 2021
> > >
> > > [2]. X-RefSeg3D: Enhancing Referring 3D Instance Segmentation via Structured Cross-Modal Graph Neural Networks, AAAI 2024

---

> > > > ### Author Response · Authors · 2024-08-14
> > > > **Response to Reviewer 6zjS**
> > > >
> > > > Dear reviewer,
> > > >
> > > > It appears that no email reminders have been sent since our last reply. In our most recent responses, we have addressed the concerns and questions you raised. Should you have further inquiries or suggestions, we would be pleased to engage in further discussion.

---

### Official Review · Reviewer_NB5W · 2024-07-12

**Soundness:** 3
**Presentation:** 3
**Contribution:** 2
**Rating:** 5
**Confidence:** 4

**Summary:**

This paper addresses the problem of referring 3D Instance Segmentation, which segments all points belonging to an object in a 3D point cloud described by a query prompt. Previous methods use a two-stage approach requiring both instance and semantic labels, which is labor-intensive. The authors propose LESS (Label-Efficient and Single-Stage), a new pipeline that only requires binary mask supervision, reducing annotation costs. Key innovations include a Point-Word Cross-Modal Alignment module, a Query Mask Predictor module, a Sentence Query Alignment module, and two new losses for weaker supervision. LESS significantly outperforms existing methods on the ScanRefer dataset.

**Strengths:**

1. The paper is clear and easy to understand. The proposed method is straightforward and well-explained
2. Strong quantitative results are reported on the ScanRefer dataset.

**Weaknesses:**

1. No substantial novelty in module design or architecture, they are already proposed in other contexts:
   + Query Mask Predictor: similar to Mask2Former: M2F3D: Mask2Former for 3D Instance Segmentation
   + Point-Word Cross-Modal Alignment: similar to cross-attention
   + Query-Sentence Alignment: cosine CLIP score
2. The paper lacks explanations of why TGNN and X-RefSeg perform worse, and in which case these methods fail.
3. The area regularization loss brings the most significant gain, but is not explored in detail, i.e. ablation of the loss coefficient.
4. Running time compared to the two-stage approaches? If it is not significantly reduced, there is no advantage compared to the two-stage approaches.
5. The results with Acc@0.5 are worse than those of TGNN and X-RefSeg (Table 1) while the results with Acc@0.25 are opposite indicating that the proposed method is better locate coarse location rather than exact matching as in TGNN and X-RefSeg.

Comment: TGNN and X-RefSeg use a pretrained 3DIS network, when training on this Referring 3DIS task, only need binary mask supervision.

**Questions:**

1. How does the loss coefficient affect the results?
2. What is the impact if there are less than 20 queries in Table 5?
3. Are there any benchmarks other than ScanRefer? The main paper only shows the result on one benchmark, which may be not sufficient. How about Sr3D and Nr3D

**Limitations:**

Yes, this paper has a session discussing its limitations.

---

> ### Author Rebuttal · Authors · 2024-08-07
>
> Thanks for your detailed and constructive comments. Next we address to your concerns as follows:
> ## Q1: For the novelty of the module design
> The core motivation of our approach is to explore a novel single-stage R3DIS to fully embrace semantic concepts from text query and visual cues into a unified paradigm, while leveraging an efficient binary supervision. Most of existing works focus on two-stage pipeline to reach R3DIS: first instance proposal network and then a new network to classify these proposal into the corresponding text query, which separate the interaction between text query and visual cues. To investigate label-efficient and single-stage pipeline in R3DIS task, we delve into fusion mechanism for better aligning visual features with textual embeddings within a cross-attention manner. Specially, we design coarse-grained feature alignment and fine-grained feature alignment strategies, corresponding to our QMP and SQA, and PWCA, respectively. Besides, the utility of area regularization and point-to-point contrastive learning are effective to eliminate ambiguities among multiple objects and backgrounds, since single-stage approach possibly struggles with positive false and negative true samples without a well-trained proposal generator network. All of the proposed components in our paper lead to our competitive SoTA and pioneering single-stage R3DIS LESS. Moreover, our methods needs only binary mask annotations to train the model while the previous ones need heavy both instance labels and semantic labels, which strikes to label-efficient and single-stage approach for this task.
> ## Q2: For the explanations of why TGNN and X-RefSeg perform worse and the cases they fail
> Firstly, as mentioned in line 31-34 in the paper, owing to accuracy of mode and the lacking of linguistic guidance, TGNN and X-RefSeg  may miss the target in the pre-segmentation stage. In this case, it is impossible to provide language-related high-quality instance proposals for the second stage. Besides, compared to our fine-grained and coarse-grained alignment, their visual-language alignment is weak, usually applying language guidance at the second stage only. We deduce these are the main reasons why both of them performed poorly. Moreover, if the targets are severely missed in the first stage or queries are complex, TGNN and X-RefSeg will perform worse.
> ## Q3: For the loss coefficient of area regularization
> We conduct further ablations in Q7 together.
> ## Q4: For the running time compared to pervious two-stage approaches
> We also conduct comparisons to address this concern, please kindly refer to the table of our **General Response to all Reviewers**.
> ## Q5: For the results with Acc@0.5 are worse than those of TGNN and X-RefSeg while the results with Acc@0.25 are opposite
> Since TGNN and X-RefSeg are all two-stage methods, that their 3D instance semantic segmentation networks in the first stage are first well-trained by both instance labels and semantic labels to support higher precision instance proposals for the second stage than ours. This benefits for the Acc@0.5 metric. However, as shown in Q7, we can still find that our LESS also surpasses both TGNN and X-RefSeg in Acc@50, which indicates our model is better on not only locating coarse location but also exact matching, thanks to our elaborate alignment strategies for training between language information and visual cues.
> ## Q6: For the comment of detailed setting of TGNN and X-RefSeg
> In our main paper for the R3DIS task, TGNN and X-RefSeg both leverage a two-stage paradigm, which consists of a 3D instance semantic network and a instance matching network. Obviously, TGNN and X-RefSeg rely on instance labels, semantic labels and binary masks to finalize referring 3DIS task rather than "only binary masks", the efficient way we used. Noting that, we trained 3D spares U-Net’s architecture from scratch without its pretrained weights, which is different from previous methods.
> ## Q7: For how does the loss coefficient affect the result
> We further conduct more ablation studies of λp2p, λseg and λarea. It can be seen that λp2p presents marginal influence in terms of final performance and we tend to set with default 0.05. And reducing λseg is advantageous for obtaining decent performance. We analyze that this is due to the reduction of λseg has the effect of amplifying the impact of the area regularization loss, thereby facilitating a stronger discrimination between backgrounds and foregrounds, leading to local mask predictions. We still observe improvements when λarea increases, especially for Acc@50 metric, which indicates our area regularization loss can effectively exclude backgrounds and get more precise masks.  We have uploaded the corresponding results on the **PDF**, please kindly refer to that due to the Rebuttal Words Limitation.
> ## Q8: For the ablation studies of the number of queries
> We further conduct more ablation studies in terms of the number of queries on ScanRefer. As shown in the following table, we ablate the number of queries as {5, 15, 20}, it can be observed that keeping more queries brings higher accuracy while lower accuracy when using fewer queries. We suppose that fewer queries can not help to learn comprehensive feature patterns while an appropriate number of queries is enough to cover the needed feature patterns.
> | Num of queries |   mIoU    |  Acc@25   |  Acc@50   |
> | :------------: | :-------: | :-------: | :-------: |
> |       5        |   32.75   |   51.81   |   28.68   |
> |       15       |   33.27   |   52.46   |   28.97   |
> | 20 (Baseline)  | **33.74** | **53.23** | **29.88** |
> ## Q9: For the validation on other dataset
> We also conduct the experiments on the Nr3d and Sr3d datasets to address this concern, please kindly refer to the table of our **General Response to all Reviewers**.
>
> **Thanks again for these constructive comments, we will take all these experimental results into consideration into our revised version.**

---

> > ### Comment · Reviewer_NB5W · 2024-08-10
> >
> > Thank you for your detailed response. However, my concerns about the novelty of your work remain unaddressed, particularly regarding how your proposed modules differ from previous methods. I was looking for a clearer explanation of how your approach offers significant advancements rather than just adopting existing techniques. Therefore, I will maintain my original rating.

---

> > > ### Author Response · Authors · 2024-08-12
> > > **Response to Reviewer NB5W**
> > >
> > > Thank you for your response! We are grateful and pleased that our response can address most of your concerns. Regarding the novelty of our work, we would like to answer from the following points and discuss the differences from previous works.
> > >
> > > - The novelty of the Query Mask Predictor and Query-Sentence Alignment. As we explained in the paper, QMP can provide candidates of the target in the scene. However, different candidates have different importance to the target and most of them contains unrelated background points. To address this issue, we propose the QSA module to weight the mask and query by calculating the relationship between the candidate and the query through the cosine similarity. Moreover, **different from CLIP[1]** use cosine, which selects the mask with the highest score as the final result, our final mask prediction is facilitated by a weighted sum of the similarity score and mask prediction. Using a weighted sum enables a soft decision-making process rather than a hard and binary choice. This can help in cases where multiple masks have similar scores, allowing the final prediction to be an aggregation of these maks, which can be more accurate than any single one. As shown in Tab.x, we also conduct related experiments about this, which also confirm the advantages of our method.   **In comparison to Mask3D [2]**, our QMP and SQA generate possible mask proposals guided by the language caption, whereas Mask3D's decoder enumerates all possible proposals in a scene without any guidance.
> > >
> > > |          |   mIoU    |  Acc@25   |  Acc@50   |
> > > | :------: | :-------: | :-------: | :-------: |
> > > |  Top-1   |   33.18   |   52.93   |   28.65   |
> > > | Baseline | **33.74** | **53.23** | **29.88** |
> > >
> > > - The novelty of the Point-Word Cross-Modal Alignment. The PWCA is based on the cross-attention structure, which is a prevalent approach in cross-modal fusion networks. **In comparison to the original cross attention model[3]**, we have removed the position encoding in order to address the disorder of 3D point clouds. Furthermore, we have incorporated an MLP Tanh gate on top of the original model in order to regulate the information flow between language and visual features.
> > >
> > > - The novelty of module design.  In contrast to previous research, which solely aligned word features with proposal features, we introduce a novel one-stage coarse-to-fine point cloud language cross-modal alignment paradigm. PWCA, QMP and QSA are design to align  word-point and sentence-mask, respectively. Furthermore, another significant innovation is the introduction of a coarse-to-fine background point filtering strategy, comprising Area Regularization Loss and Point-to-Point Contrastive Loss, which enables the effective processing of high-complexity, large-scale 3D scenes.
> > >
> > > - Finally, related experiments shown that compared to previous works, our LESS is simple yet effective not only in SoTA performance but also in efficient time consumption with only binary mask supervision. As mentioned in conclusion of our paper, we aims to construct a straightforward and universal single-stage baseline to establish a solid foundation for this task, rather than focusing solely on the performance enhancement offered by a single novel model architecture, which sheds new lights on future exploration on how to better align the visual feature with language feature (cross-modal) or propose different training strategies.
> > >
> > > We would like to express our gratitude for your valuable suggestions and for highlighting the existing issues. Hope that our response could provide the necessary clarification to address your concerns. If you have any additional questions or suggestions, we are very glad to have a further discussion.
> > >
> > > [1]. Learning Transferable Visual Models From Natural Language Supervision, ICML 2021.
> > >
> > > [2]. Mask3d: Mask transformer for 3d semantic instance segmentation, ICRA 2023.
> > >
> > > [3]. Attention is All You Need, NeurIPS 2017.

---

> > > > ### Comment · Reviewer_NB5W · 2024-08-12
> > > >
> > > > It seems there may have been a misunderstanding. I was expecting a detailed explanation of the significant differences between the proposed modules and existing techniques. Specifically:
> > > > * Query Mask Predictor: How does it differ from the approach used in Mask2Former, specifically in M2F3D for 3D Instance Segmentation?
> > > > * Point-Word Cross-Modal Alignment: This appears similar to cross-attention. Could you clarify the distinctions?
> > > > * Query-Sentence Alignment: How does your method compare to using the cosine CLIP score?

---

> > > > > ### Author Response · Authors · 2024-08-13
> > > > > **Response to Reviewer NB5W**
> > > > >
> > > > > Here are the differences. We will point them out one by one:
> > > > >
> > > > > - **Query Mask Predictor:** We adopted an architecture similar to Mask3D [1] and Mask2Former [2], but since our single-stage model does not require semantic labels for supervision, we removed the semantic branch; And since only specific masks need to be predicted, the process of complex Hungarian Matching  can be omitted.
> > > > >
> > > > > - **Point-Word Cross-Modal Alignment:** The distinctions are we remove the position encoding in order to address the disorder property of 3D point clouds, and we incorporate an MLP Tanh gate following the cross attention to regulate the information flow between language and visual features.
> > > > >
> > > > > - **Query-Sentence Alignment:** We use cosine similarity to calculate the similarity score R between sentence features and object queries. The calculation formula is provided in equation (5) of the paper. Cosine similarity is a technique widely used in common machine learning to measure the similarity between two vectors, in particularly for cross-modal vectors. Besides, similarity calculation is only a part of Query-Sentence Alignment, and the rest involves weighted summation to select the final prediction mask.
> > > > >
> > > > > In our previous response, we provided a detailed introduction of our design concept about the entire framework and the loss function, which demonstrates the effectiveness and efficacy of our proposed single-stage pipeline, and superb even SoTA performances can be obtained within an label-efficient training. We do believe this line of works will raise increaing attentions to explore better training strategies or model architectures based on single-stage pipeline, instead of separate cross-modal interaction and multi-stage training stages.
> > > > >
> > > > > [1]. Mask3d: Mask transformer for 3d semantic instance segmentation, ICRA 2023.
> > > > >
> > > > > [2]. Masked-attention Mask Transformer for Universal Image Segmentation, CVPR, 2022.

---

> > > > > > ### Author Response · Authors · 2024-08-14
> > > > > > **Response to Reviewer NB5W**
> > > > > >
> > > > > > Dear reviewer,
> > > > > >
> > > > > > It appears that no email reminders have been sent since our last reply. In our most recent responses, we have addressed the concerns and questions you raised. Should you have further inquiries or suggestions, we would be pleased to engage in further discussion.

---

> > > > > > > ### Comment · Reviewer_NB5W · 2024-08-14
> > > > > > >
> > > > > > > Thank you for your response. However, based on your clarification, I still don’t see a significant contribution. The differences appear to be mainly in the implementation details, with only minor modifications to make it work.

---

> ### Author Response · Authors · 2024-08-14
> **Response to Reviewer NB5W**
>
> Thank you for your questions and suggestions. A recent work, OpenScene[1] has established a straightforward and efficient baseline network in the field of open vocabulary 3D segmentation using a simple cross modal distillation method based on primary cosine smilarity driven, but also with a pioneering explroation. The objective of our work is to present a straightforward and effective single-stage baseline for the task of 3D referring segmentation. In addition to a single stage architecture design, our work makes a further contribution in the form of a single-stage coarse-to-fine cross-modal alignment network and a coarse-to-fine background point filtering loss function. These have been developed with the aim of achieving state-of-the-art performance under efficient binary mask supervision, which have been demonstrated with extensive ablations mentioned before.
>
> Thank you again for your suggestions. We sincerely hope that you will be able to evaluate the contribution of our work based on its overall design and the differences that it presents in comparison to previous works [2][3], and also the hitting cornerstone of single-stage pipeline of this task.
>
> [1]. Openscene: 3d scene understanding with open vocabularies, CVPR 2023.
>
> [2]. Text-guided graph neural networks for referring 3d instance segmentation, AAAI 2021
>
> [3]. X-RefSeg3D: Enhancing Referring 3D Instance Segmentation via Structured Cross-Modal Graph Neural Networks, AAAI 2024

---

### Author Rebuttal · Authors · 2024-08-07

# General Response to all Reviewers:

Dear all Reviewers,

We would like to express our great thank and appreciation to all the reviewers for their constructive and thoughtful comments on our submission. In this rebuttal response, we address the questions raised by the reviewers and clarify how we will revise our manuscript in repsonse to the offered comments. Our paper aims at proposing a novel single-stage Referring 3D Instance Segmentation to fully embrace semantic concepts from text query and visual cues into a unified paradigm, which differs from most exisitng two-stage works necessitating both instance and semantic labels for each object first and using a separate interaction between text and visual cues during the training. We are motivated and glad that our proposed method is well-explained, appealing and exhibiting a certain level of novelty (Reviewer NB5W, 6zjS, pLC3 and Monv), is of clear writing and structure for understanding (Reviewer NB5W, 6zjS, pLC3 and Monv), demonstrates competitive SoTA performance on ScanRefer benchmark (Reviewer NB5W, 6zjS, pLC3 and Monv) and sheding meaningful research for the future to enable label-efficient learning for this community (Reviewer pLC3 and Monv).

Here we answer some general questions of reviewers:

## Q1. For the evaluation on more benchmarks, especially on Nr3D and Sr3D
Thanks for the constructive suggestions to conduct more validations on other benchmarks.

We further employ our method on extra Nr3D and Sr3D as shown in table below. Following the evaluation setting in TGNN, we can observe that the performances of our LESS consistently surpasses the performances of TGNN by 3.6% and 1.8% on Nr3d and Sr3d, respectively. This further demonstrates the effiectiveness of our method, still achieving competitive performances on additional Nr3D and Sr3D, which presents promising potential for the future works. We will add these new experimental results to our revised manuscript.

|      | Method | Overall  |   Easy   |   Hard   | View-dep. | View-indep. |
| :--: | :----: | :------: | :------: | :------: | :-------: | :---------: |
| Nr3D |  TGNN  |   37.3   |   44.2   |   30.6   |   35.8    |    38.0     |
|      |  Ours  | **40.9** | **47.4** | **35.2** | **39.7**  |  **41.9**   |
| Sr3D |  TGNN  |   45.0   |   48.5   |   36.9   |   45.8    |    45.0     |
|      |  Ours  | **46.8** | **50.5** | **37.8** | **46.6**  |  **46.3**   |

## Q2. For the comparisons with pervious two-stage approaches, especially with TGNN and X-RefSeg

Thanks for the constructive suggestions regarding more comprehensive comparisons with two-stage methods.

Firstly, we compare the training and inference time of our method with both TGNN and X-RefSeg, shown in the following table. All comparative experiments are conducted on a NVIDIA 4090 GPU machine and the batch sizes of three methods are kept the same. For the two-stage training and inference of TGNN and X-RefSeg, we strictly follow the settings in their open source codes. From the table, we can find that, our LESS consumes less time than both of TGNN and X-RefSeg in terms of training and inference cost, which showcases the promising potential of single-stage approach, our proposed LESS.


|  Method  | Inference (Whole Dataset) (min) | Inference (Per Scan) (ms) | Training (Stage 1) (h) | Training (Stage 2) (h) | Training (All) (h) |
| :------: | :-----------------------------: | :-----------------------: | :--------------------: | :--------------------: | :----------------: |
|   TGNN   |              27.98              |          176.57           |         156.02         |          8.53          |       164.55       |
| X-RefSeg |              20.00              |          126.23           |         156.02         |          7.59          |       163.61       |
|   Ours   |            **7.09**             |         **44.76**         |           -            |           -            |     **40.89**      |


Secondly, we reimplement the TGNN and X-RefSeg with the same RoBERTa backbone as ours to conduct experiments on ScanRefer benchmark, since these exisitng works have not applied RoBERTa backbone as language module for their experiments. We then follow their settings to train and validate the performances. As shown in table below, we can obtain their newly corresponding results, 28.83% mIoU for TGNN (RoBERTa) and 30.72% mIoU for X-RefSeg (RoBERTa), which still shows inferior performance to ours 33.74%, which further validates the superiority of our single-stage prardigm.

|               Method               |   mIoU    |  Acc@25   |  Acc@50   |
| :--------------------------------: | :-------: | :-------: | :-------: |
|             TGNN (GRU)             |   26.10   |   35.00   |   29.00   |
|            TGNN (BERT)             |   27.80   |   37.50   |   31.40   |
|           TGNN (RoBERTa)           |   28.83   |   39.15   |   32.50   |
|           X-RefSeg (GRU)           |   29.77   |   39.85   |   33.52   |
|          X-RefSeg (BERT)           |   29.94   |   40.33   |   33.77   |
|         X-RefSeg (RoBERTa)         |   30.72   |   41.54   |  **34.42**   |
| Ours (RoBERTa)   |   **33.74**   | **53.23** |   29.88   |


Overall, we can find that our approach can not only exhibits much more efficient training/inference cost but also superior performances when equiping TGNN and X-RefSeg with the same RoBERTa backbone. Thanks for the thoughful comments from all reviewers, we will take these comparisons into consideration by adding these experimental results in our revised version. Next, for the specific questions of each kind reviewer, we answer them in the reply to each one.

The following PDF document contains tables for three ablation experiments for Reviewer NB5W, the inclusion of which is due to space limitations.

---

### Decision · Program_Chairs · 2024-09-25

**Decision:**

Accept (poster)

**Comment:**

The reviewers all agree on the proposed label efficient single stage 3d referring expression modeling being either reasonable and straightforward or a sufficiently novel contribution. The rebuttal and discussion clarified some question about evaluation and setup, and overall the authors are encouraged to take into account the reviewer comments.